# Signatures of Admixture and Genetic Uniqueness in the Autochthonous Greek Black Pig Breed Deduced from Gene Polymorphisms Affecting Domestication-Derived Traits

**DOI:** 10.3390/ani13111763

**Published:** 2023-05-26

**Authors:** Anisa Ribani, Valeria Taurisano, Despoina Karatosidi, Giuseppina Schiavo, Samuele Bovo, Francesca Bertolini, Luca Fontanesi

**Affiliations:** 1Department of Agricultural and Food Sciences, Division of Animal Sciences, University of Bologna, Viale Giuseppe Fanin 46, 40127 Bologna, Italy; anisa.ribani2@unibo.it (A.R.); valeria.taurisano2@unibo.it (V.T.); giuseppina.schiavo2@unibo.it (G.S.); samuele.bovo@unibo.it (S.B.); francesca.bertolini3@unibo.it (F.B.); 2Research Institute of Animal Science, General Directorate of Hellenic Agricultural Organisation “Demeter”, Paralimni Giannitsa, 58100 Pella, Greece; despinakaratosidi@elgo.gr

**Keywords:** animal genetic resource, DNA markers, introgression, *MC1R*, population genetics, SNP, *Sus scrofa*

## Abstract

**Simple Summary:**

Autochthonous pig breeds are important genetic resources, well adapted to local climatic conditions, environments, and traditional production systems, where they are associated with local and niche markets. The Greek Black Pig breed is the only local pig breed recognized in Greece. In this study, we started a population genetic characterization of this breed by analyzing a few gene markers associated with morphological and production traits and that usually differentiate wild boars from domestic breeds. The obtained results showed that, in the past, this breed experienced genetic admixture from two sources, wild boars and cosmopolitan breeds. On the one hand, this situation might raise some concerns for the genetic integrity of this animal genetic resource. On the other hand, this might contribute to within-population genetic variability reducing the problem of inbreeding of the small breed population. In this breed, we also identified a novel allele in the melanocortin 1 receptor (*MC1R*) gene, resulting in a new hypothesis on the function of the encoded protein in regulating the cascade signals and leading to the production of different pigmentation. This result showed that local untapped breeds can be the reservoir of interesting genetic variants useful to better understanding underlying basic biological functions in mammals.

**Abstract:**

The Greek Black Pig (or Greek Pig) is the only recognized autochthonous pig breed raised in Greece, usually in extensive or semi-extensive production systems. According to its name, the characteristic breed coat color is solid black. In this study, with the aim to start a systematic genetic characterization of the Greek Black Pig breed, we investigated polymorphisms in major genes well known to affect exterior and production traits (*MC1R*, *KIT*, *NR6A1*, *VRTN* and *IGF2*) and compared these data with population genetic information available in other Mediterranean and Western Balkan pig breeds and wild boars. None of the investigated gene markers were fixed for one allele, suggesting that, in the past, this breed experienced introgression from wild boars and admixture from cosmopolitan pig breeds, enriching the breed genetic pool that should be further investigated to design appropriate conservation genetic strategies. We identified a new *MC1R* allele, containing two missense mutations already reported in two other independent alleles, but here present in the same haplotype. This allele might be useful to disclose biological information that can lead to better understanding the cascade transmission of signals to produce melanin pigments. This study demonstrated that autochthonous genetic resources can be an interesting reservoir of unexpected genetic variants.

## 1. Introduction

Since the first domestication events that occurred in *Sus scrofa*, artificial directional selection in pigs has been the main driving force that substantially shaped the domestic genetic pools and differentiated them from the original European and Asian wild boar genetic background [1,2,3,4]. Then, many breeds and populations were derived from this process, including the modern cosmopolitan and highly selected breeds and lines, which largely substituted autochthonous less productive breeds, becoming the dominant genetics of the intensive production systems [2]. Autochthonous pig breeds, however, still represent important genetic resources in many countries, well adapted to local climatic conditions, environments, and traditional production systems, where they are associated with local niche markets [5,6]. Their pork products are usually known for their high quality, mainly due to the high intermuscular and intramuscular fat content [7,8].

Native pig populations are also present in many European countries of the Mediterranean region, where they are well adapted to hot, dry and harsh climatic conditions and poor pastures [5]. These populations, in several cases, are still unexplored, although recent research efforts have genetically characterized some local breeds of this area [5,9,10,11,12,13].

The Greek Black Pig (also known as Greek Pig) is an autochthonous breed raised in Greece. Its origin dates back to the ancient time even if it is clear that its genetic pool might have been shaped by many more recent events [14]. The Greek Black Pig breed has been the only pig breed raised in Greece until 1960 where it constituted one of the main components of the traditional extensive outdoor livestock production sectors in the country [15]. These pigs are considered well adapted to the environmental conditions of the Greek mainland and Aegean islands [16]. The relevance of this breed, however, decreased with the advent of intensive pork production systems, where cosmopolitan improved breeds and lines substituted the Greek Black Pig population [7]. Nowadays, there are small nuclei of Greek Black Pigs spotted in all of continental Greece, accounting for approximately 3000 heads, mainly concentrated in the Central and North regions of the country. As it can also be deduced from the name of this breed, these animals usually have a solid black coat color (with rarely few white spots or brownish regions), with medium–long hairs (Figure 1).

Sows reach reproductive maturity after eight months of age and produce two litters per year, with approximately eight piglets per litter that have a weaning rate of approximately 80% at birth. The pigs are reared in a semi-extensive farming system, based on mountain pastures, where they are fed with acorns, except for the mating season in which they are supplemented with concentrates [15]. The pigs grow slowly and are slaughtered when they reach 80–130 kg live weight.

Thus far, few studies have provided genetic information on this breed. Michailidou et al. [16] analyzed several microsatellite markers in pigs sampled from different herds and reported a quite high level of genetic variability. Laliotis et al. [17] investigated polymorphisms in three candidate genes associated with reproduction traits in sows of this breed. Preliminary information on the distribution of melanocortin 1 receptor (*MC1R*) gene alleles in this breed was reported in another study [18].

Polymorphisms in this gene, which determine the *Extension* locus allele series, largely affect the coat color of the pigs [19,20]. Starting from the wild-type alleles (*E^+^*, also indicated as *MC1R*1*), found in wild boars, mutations produced other alleles that were selected over the domestication process of the pig [21] and that characterize many domestic breeds. The dominant *E^D1^* or *MC1R*2* alleles, originating from the Asian lineage, and the dominant *E^D2^* or *MC1R*3* allele, originating from the European lineage, determine black coat color. The red coat color observed in few breeds (e.g., Duroc) is determined by the homozygous genotype of the recessive *e* (*MC1R*4*) allele. The black spotted coat color is associated with the *E^P^* allele, also known as *MC1R*6*. The nomenclature of the allele series of the *MC1R* allele has also been defined following a four-digit rule, whose correspondence with other nomenclatures has recently been summarized [22]. For example, *0301* indicates allele *E^D2^* and *0401* indicates allele *e*, where the first two digits define the main effect related to the evolutionary lineage and the last two digits define the specific ID of the allele within the same evolutionary group [21].

Polymorphisms in another major gene for coat color in pigs (KIT proto-oncogene, receptor tyrosine kinase or *KIT*) explain the allele series at the *Dominant white* (*I*) locus [22]. The dominant white coat color is determined by a combination of copy number variations (CNVs) and a splice mutation at intron 17 of the *KIT* gene [23,24,25,26]. The belted phenotype, which characterizes some belted breeds such as the Italian autochthonous Cinta Senese breed, is associated with the g.8:43597545C > T mutation in the *KIT* gene [27].

In addition to the coat color, the domestication process in the pig has acted to shape other morphological traits, including the increased length of the animals determined by a larger number of vertebrae and an increased number of teats. These correlated traits are considered quantitative traits that are affected by many genes, including the variability in two major genes: a missense mutation (g.1:265347265T > C or p.P192L) in the nuclear receptor subfamily 6 group A member 1 (*NR6A1*) gene and polymorphisms in the vertnin (*VRTN*) gene [28,29]. The mutated alleles are associated with increased number of vertebrae and teats and increased length of the animals compared to the wild boar ancestral alleles [2].

Growth and carcass parameters are other quantitative traits affected by the variability of many genes, including a major gene: a single-nucleotide polymorphism in the intron 3 (g.3072G > A) of the insulin-like growth factor-2 (*IGF2*) gene causes relevant imprinted effects on muscle and fat deposition as well as on growth of the pigs [30,31]. Selection for increased muscle mass and improved growth rate leads to a shift in the allele frequencies in the population toward the mutated (A) allele [32].

Markers in genes that determine domestication-derived traits or that affect economic relevant traits have been suggested to be more useful for the purpose of genetic characterization of untapped breeds and for the development of conservation programs aimed to maintain distinctive specific characteristics and genetic features of local genetic resources [9,13]. This is particularly relevant in autochthonous pig breeds that are raised in outdoor systems and that might frequently experience close contacts and genetic introgression from wild boars and/or other pig populations [9,13,33]. Genetic information at these polymorphic sites can also be useful to reconstruct the genetic history of local pig breeds [9,33].

In this study, with the aim to start a systematic genetic characterization of the Greek Black Pig breed, we investigated polymorphisms in major genes (known to affect exterior and production traits: *MC1R*, *KIT*, *NR6A1*, *VRTN* and *IGF2*) in pigs of this breed and compared these data with population genetic information at the same polymorphic sites available in other European pig breeds and wild boars. We identified a new *MC1R* allele that might shed light on the effect of polymorphic sites in the function of the encoded protein and obtained population genetic information useful to understand the genetic events that contributed to shape the Greek Black Pig breed genetic background.

## 2. Materials and Methods

### 2.1. Greek Black Pig Samples

A total of 59 pigs (seven boars and 52 sows) belonging to the Greek Black Pig breed were sampled in the Thessaly region (Greece, GPS coordinates: Latitude 300979,367; Longitude 4395953,322). Selection of animals for sampling was performed avoiding highly related animals (no full- or half-sibs) according to the information provided by the farmers, even if registered pedigree information was not available, and including only adult individuals that had adult morphology. All animals had a solid black coat color with the typical morphological traits of the breed.

### 2.2. DNA Extraction and Genotyping of Polymorphisms in the Greek Black Pigs

Genomic DNA was extracted from hair roots, collected from the sampled Greek Black Pigs, using standard phenol-chloroform-isoamyl alcohol DNA extraction protocol [34].

A total of nine autosomal polymorphisms in five genes (*MC1R*, *KIT*, *NR6A1*, *VRTN* and *IGF2*) were genotyped: three single-nucleotide polymorphisms (SNPs) and one insertion/deletion (indel) of two nucleotides (c.67insCC) in the *MC1R* gene whose combination can discriminate all major alleles at this gene (*MC1R*1*, *MC1R*2*, *MC1R*3*, *MC1R*4* and *MC1R*6*) [19,20]; the duplication breakpoint of the *KIT* gene, which distinguishes the *Dominant white* alleles determined by CNVs (alleles *I*) from the other alleles [22,35], and in the same gene the g.43597545T > C SNP, which is associated with the belted phenotype [27]; one missense mutation in the *NR6A1* gene (p.P192L or g.299084751C > T) which is the causative mutation of the QTL identified on porcine chromosome (SSC) 1 affecting the number of vertebrae and teats [28]; the indel 20311_20312ins291 in the *VRTN* gene, determined by a SINE insertion or deletion, which distinguishes the two alleles (the mutated allele Q, with the insertion, from the wild-type allele q, without the insertion) of the QTL identified on SSC7 affecting the number of vertebrae and teats [29]; the g.3072G > A SNP in intron 3 of *IGF2* gene, that has the mutated allele A associated with increased muscle mass deposition and growth performances [30]. All SNPs were analyzed using a PCR-RFLP method. The indels and the duplication breakpoint were analyzed using PCR fragment analyses. Sanger sequencing was also applied to confirm *MC1R* genotyping data from all Greek Black pigs and the genotyping results of a few animals for the other genes, as previously described [26,27]. PCR primers and detailed genotyping protocols derived from [13,26,27,32,33] and from the sequencing are summarized in Appendix A.

### 2.3. Genotyping Data from Other Pig Populations

Genotyping data of an additional 22 pig breeds and one wild boar population for a total of 831 animals were retrieved from previous studies [9,13,26,27,32,33,36] or by mining whole-genome sequencing of DNA pools obtained by a previous study [11]. Whole-genome sequencing datasets were analyzed using the pipeline described in [11,12]. Pig breeds were from Portugal (Alentejana and Bísara), Spain (Majorcan Black), France (Basque and Gascon), Germany (Schwäbisch Hällisches Schwein), Italy (autochthonous breeds: Apulo Calabrese, Casertana, Cinta Senese, Mora Romagnola, Nero Siciliano and Sarda; cosmopolitan breeds: Italian Duroc, Italian Landrace and Italian Large White), Lithuania (Lithuanian Indigenous Wattle and Lithuanian White Old Type), Slovenia (Krškopolje), Croatia (Black Slavonian and Turopolje), and Serbia (Moravka and Swallow Bellied Mangalitsa) [9,13,26,27,32,33]. The wild boar population included boars from the Italian peninsula [13,36]. A summary of the genotyping information for these breeds and populations, including the number of samples per breed and source of the information (previous studies or mining whole-genome sequencing datasets carried out in this study), is reported in Appendix A.

### 2.4. Data Analyses

Allele and genotype frequencies obtained in the Greek Black Pig breed were used to evaluated if markers were in the Hardy–Weinberg equilibrium computed with the HWE software program (Linkage Utility Programs, Rockefeller University, New York, NY, USA).

A median-joining network of porcine *MC1R* alleles, including the new allele identified in this study, was manually curated based on the network reported by previous studies [21,37].

Principal Component Analysis (PCA) based on allele frequencies was used to evaluate the relationships among all investigated pig populations, including the Greek Black Pig breed. PCA analysis was carried out in R v.3.4.4 after having computed a dissimilarity matrix D, where each value represents the Euclidean distance *d* between pairs of populations. Cluster representation of the analyzed populations has been generated with the ‘dist’ and ‘hclust’ functions of R v.3.4.4 using allele frequency information to calculate the Euclidean distance among groups.

## 3. Results

### 3.1. Genotyping Results of Greek Black Pigs

Allele frequencies obtained in the Greek Black Pigs for the five investigated genes are summarized in Table 1. All genes were polymorphic in this breed.

#### 3.1.1. Polymorphisms in the *MC1R* Gene

In the *MC1R* gene, allele *E^D2^* (*MC1R*3*), that determines a dominant black phenotype, was the most frequent allele. However, this allele was not fixed in the Greek Black Pig breed, as it would be expected from the coat color of the analyzed pigs. The other five alleles were identified even if at lower frequencies; four of which were already described in other pig breeds and populations: *E^+^* or *MC1R*1*, which should be derived from wild boars, was identified in nine pigs but always in heterozygous state with *E^D2^*; *E^D1^* (*MC1R*2*), the dominant black allele of Asian origin, was the rarest allele (identified only in one pig and in heterozygous state with the other dominant black allele); *e* (*MC1R*4*), the recessive red allele, was identified in 18 pigs but always in heterozygous state with the dominant black allele of European origin; *E^P^* (*MC1R*6*) was identified only in two pigs, in heterozygous state with the *E^D2^*allele. A new allele that, as far as we know, has not been identified in any other breeds thus far, was detected with a frequency of 4.2%: two pigs were homozygous for this allele, and one pig was heterozygous with *E^D2^*. We named this new allele as *E^D2e^*, according to its putative phenotypic effect, as all pigs carrying this allele (in particular, the homozygous pigs) were completely black. This allele is derived by the combination of two missense mutations already reported, but separately, in other alleles, which produce a new haplotype: c.370G > A (rs326921593, which leads to the p.D124N substitution), already described to characterize the dominant black *MC1R*3* allele (allele *E^D2^*), indicated also as *0301*; c.727G > A (rs321432333, which leads to the p.A243T substitution), identified in the recessive *MC1R*4* allele (allele *e*). Sequence alignment of this new allele, allele *E^D2e^*, with alleles *E^D2^* and *e* is reported in Figure 2a. At another informative site [c.491C >T, p.A164V; that distinguishes allele *e* (*MC1R*4*), that has the mutated nucleotide T at this position), from all other alleles that have nucleotide C at the same position] [19,21,22], this new *MC1R* haplotype carries the same nucleotide of the *E^D2^* (*MC1R*3*) allele, that means it carries nucleotide C (Figure 2a). This new *MC1R* pig sequence has been deposited in the European Nucleotide Archive (ENA) with accession number PRJEB59417. According to the two other nomenclatures of the allele series at this gene [19,20,21,22,37,38], and following the progressive number of this series reported in previous studies [19,20,21,22,37,38,39], we therefore named this new allele as *MC1R*10* and *0302* (according to one of its putative evolutionary origin, and effect on coat color, following what previously reported [21,37]; see also Section 4). Figure 2b reports the phylogenetic tree including different *MC1R* alleles, derived from Fang et al. [21] and Linderholmet et al. [37], where we added this new *MC1R* allele (*0302*, here also indicated as *E^D2e^* and *MC1R*10*). The reticulate position of this new *MC1R* sequence in the phylogenetic tree is due to the uncertainty of its origin between two equally plausible hypotheses: a de novo but recurrent mutation (c.727G > A, which leads to the p.A243T substitution) that occurred in an *E^D2^* haplotype sequence; a recombination between an *E^D2^* allele and an *e* allele, which constituted a new haplotype carrying two mutations (p.N124D and p.A243T) of the two original alleles (Figure 2a; see Section 4 for more details).

#### 3.1.2. Polymorphisms in the Other Genes

Only one pig carried the duplication of the *KIT* gene, that is associated with the *Dominant White* (*I*) allele, which is common in white breeds and populations such as Large White and Landrace. The *KIT* allele associated with the belted phenotype (g.8:43597545T) was detected, in heterozygous state, only in two other pigs. All these pigs had a typical solid black coat color. All remaining pigs had the wild-type genotypes in the investigated *KIT* gene markers. The markers were in the Hardy–Weinberg equilibrium.

The two investigated genes affecting vertebral number and teat number, *NR6A1* and *VRTN*, were not fixed for the domestic (or mutated) alleles, with some differences in allele frequencies. In the *NR6A1* gene, the mutated g.265347265T allele was the prominent allele (83.9%) in the Black Greek Pig breed. The wild-type C allele was present in 11 pigs. This polymorphic site was not in the Hardy–Weinberg equilibrium due to the lack of heterozygous animals (TT = *n*. 48; CT = *n*. 3; CC = *n*. 8). The analyzed polymorphism in *VRTN* (the SINE insertion or deletion) showed a balanced allele frequency with the mutated Q allele (with the insertion) that had a frequency of 44.9% and the wild-type allele (q, without insertion) that had a frequency of 55.1%. Again, at this polymorphic site, the investigated population was not in the Hardy–Weinberg equilibrium (*p* = 0.04) due to an excess of heterozygous animals (QQ = *n*. 8, Qq = *n*. 37; qq = *n*. 14).

A prevalence of the g.3072A allele was detected for the *IGF2* gene marker (frequency: 0.856). The G allele, considered the ancestral allele [30,40,41], was present mostly in the heterozygous state, except for only one pig that carried the GG genotype. This polymorphic site was in the Hardy–Weinberg equilibrium.

### 3.2. Comparative Analyses with Other Pig Breeds and Populations

Allele frequencies for the same markers analyzed above were obtained for other 22 pig breeds and one wild boar population using whole-genome sequencing data from DNA pools [11]. Appendix A reports the obtained allele frequency information that was merged with the related information reported above for the Greek Black Pig breed to produce a PCA plot (Figure 3a) and a cluster dendogram (Figure 3b) including all analyzed pig breeds and the wild boar population. The limited number of markers used in these analyses made it possible to obtain a preliminary representation of the genetic distance between the Greek Black Pig breed and all other investigated European pig genetic resources. Regardless these limits, it was interesting to note that the Greek Black Pig breed clustered with one of the geographically closest breeds included in this study: the Italian Apulo-Calabrese breed, that is raised in the Central-Southern regions of the Italian peninsula. Other genetically close breeds were Krškopolje (a Slovenian breed) and Schwäbisch Hällisches Schwein (a German breed), as also evidenced in both PCA and in the cluster dendrogram.

In both analyses, the Black Greek Pig breed was not closely positioned or clustered together with the wild boar population, even if wild-type alleles at all markers were also reported in this domesticated population.

For example, as also evidenced in several other autochthonous breeds raised in extensive or semi-extensive production systems, gene flow from wild boar populations might have contributed to increase genetic variability at some of the analyesed genes (Appendix A). This is particularly evident for the presence of allele *E^+^* in the *MC1R* gene and of the wild-type allele of the *NR6A1* gene in several pig breeds, including the Greek Black Pig breed. Gene flow might be also derived from other cosmopolitan breeds, again as also evidenced by the presence of several domestic alleles in the *MC1R* and *KIT* genes. This signature of reverse gene flow is not only evident in the Greek Black Pig breed but also in several other autochthonous breeds (Appendix A).

## 4. Discussion

The Greek Black Pig breed is the only autochthonous pig breed raised in Greece [42]. In this pig breed, none of the investigated gene markers, that are associated with domestication-relevant traits, were fixed for one allele. This is probably derived by the fact that several genetic events might have contributed to shape and enrich the genetic pool of this autochthonous animal genetic resource.

Natural gene flow from wild boars, due to their close contacts with this domestic population, which, in turn, is caused by the extensive or semi-extensive systems used to raise Greek Black Pigs, can explain the quite high frequency of the *MC1R E^+^* allele and *NR6A1* T allele in this breed. These two wild-type alleles are usually not present in domestic pig populations or, if present, the same gene flow mechanism is assumed [9,13,36,43]. It is not clear if this result would be due to a dedomestication or feralization process [13,44,45,46] that this breed experienced, with a reverse flow of wild-type alleles that re-entered into the breed population from which, the artificial directional selection indirectly probably worked to purge them out. These alleles are associated with wild-type morphological features and/or lower performances. This reverse process should be carefully monitored to ensure the original genetic integrity of pig autochthonous breeds, that should usually carry domestic-derived alleles. Therefore, conservation programs of local pig genetic resources might better control unwanted mating with the wild counterparts and might design breeding plan to reduce the frequency of these wild-type alleles in the breed population, where they are not part of the original domesticated genetic pool.

On the other hand, the presence of other domestic alleles, in particular at coat color genes, in addition to what would be expected according to the breed-specific black coat color, may indicate admixture between the Greek Black Pig breed and other pig domestic breeds. The presence of more than one allele at coat color loci might also explain the presence of a few non-completely black pigs in this breed population, as also indicated in some reports [42]. It is also clear, however, that this introgression could have occurred before the beginning of the conservation program of this breed that started in the year 2000. Since then, the conservation program that was established for this breed can exclude admixture with any other cosmopolitan breeds. Uncontrolled crossbreeding with highly selected and more productive breeds, with the aim to improve their performances, has been a common practice in several local pig populations before conservation programs were established [1,9,47]. This is the other face of the coin, that should be monitored and controlled to ensure or reconstruct, again, the original genetic integrity of this local breed, which can also provide a better adaptation to harsh environments.

The most interesting results that we obtained was the identification of a novel *MC1R* allele that, as far as we know, was not previously described in any other pig breeds or populations. We named this allele according to the different nomenclatures used to describe the allele series at this locus/gene: *E^D2e^*, following the nomenclature of the *Extension* locus, which indicates the black dominant effect and its origin derived from the Western European domestic lineage [19,20,22]; *MC1R*10*, following the *MC1R* gene number of alleles, as already reported in previous publications and sequences deposited in GenBank [9,19,20,22,36,37,38,39]; *0302*, following a more informative nomenclature that combine the phenotypic effect and the number of *MC1R* sequences reported for the same lineage and with the same effect [21,36]. This allele combines two missense mutations already reported in other *MC1R* alleles, with a suggested opposite functional effect when present alone: p.N124D, a missense mutation located in the third transmembrane domain of the MC1R protein (TM3), that is associated with the dominant black coat color and that is expected to produce a constitutive activation of the receptor with a downstream signal transmission, mediated by the cAMP pathway, that leads to eumelanin production in the melanocytes [19,48]; p.A243T, a missense mutation in a conserved position of the sixth transmembrane domain of the MC1R protein (TM6), suggested to disrupt the receptor function and for this reason it has been indicated to be the causative mutation of the recessive *e* allele at the *Extension* locus in pig [19]. According to the phenotypic effect of this novel allele, that is associated with the black coat color phenotype in the carrier pigs, it seems that the p.N124D amino acid change plays an intra-haplotype dominant functional effect over the second missense mutation present in the same haplotype (p.A243T) which might not have a relevant functional role in the activity of the receptor, at least in this combination. This result could potentially challenge the first interpretation of the effect of the p.A243T variant being the only causative mutation of the red coat color phenotype in Duroc pigs, deduced by the fixation of this allele in this red pig breed [19]. More detailed pharmacological characterizations, through in vitro evaluation of basal cAMP production and agonist-induced cAMP activity of the different MC1R protein variants, are needed to evaluate the specific functional effects of the described missense mutations and to support their phenotypic effect on coat color. It is worth mentioning that we recently suggested that another gene, OCA2 melanosomal transmembrane protein (*OCA2*), encoding the homolog of the mouse *p* (pink-eyed dilution) gene, located on SSC15, might play a relevant role in affecting coat color in Duroc pigs [11].

It remains to infer the way in which this novel *MC1R* allele was originated. According to the obtained *MC1R*10* nucleotide sequence, two equal probable hypotheses, as mentioned above, can be constructed. The first hypothesis assumes that a de novo and recurrent missense mutation (c.727G > A) causing the amino acid substitution p.A243T occurred in the *MC1R*3* sequence, producing a new haplotype. This hypothesis might be supported by the fact that in another informative position (c.491C > T), allele *MC1R*10* had the same nucleotide of allele *MC1R*3*. However, as the c.727G > A is downstream from the other two informative nucleotide substitutions that distinguish *E*^D2^ from *e*, it is not possible to completely exclude that *MC1R*10* could have been produced by a rare recombination event involving both *E^D2^* and *e* right in the middle between the c.491C > T and the c.727G > A polymorphic sites to obtain the new haplotype that contained both missense mutations in the same allelic form. To lead these hypotheses in one or the other direction, other informative nucleotide sites should be obtained by sequencing downstream flanking regions and reconstruct a larger haplotype structure of the new *MC1R*10* allele.

The low frequency of the *MC1R*10* allele in the breed population can also suggest that this novel allele probably emerged quite recently in the Greek Black Pig breed, through a new genetic event that could have good chances to be occurred in this breed. In this context, it is interesting to note that alleles *E^D2^* and *e* (the two alleles that contains the missense mutations that are combined in the novel *E^D2e^* (*MC1R*10*) allele, are the two most frequent alleles in the Greek Black Pig breed. Therefore, a recombination event between these two alleles (the second hypothesis on the origin of *MC1R*10*, mentioned above) would be possible in this breed. It also seems more plausible that this allele is a new allele that emerged in this breed as no other studies, as far as we know, described this form in any other pig populations (even if we cannot completely exclude the possibility that it was introgressed from another, unknown and uncharacterized, pig population).

A few other gene markers associated with economically relevant traits were analyzed in the Greek Black Pig breed, providing some preliminary information on the population genetic diversity and structure. The mutated *Q* allele in the *VRTN* gene, which has been associated with a larger number of vertebrae and teats than the wild-type allele in many pig breeds (e.g., [29,49,50]) had a balanced frequency, similar to what is observed in several other autochthonous breeds (Appendix A), suggesting that a strong selection pressure is not acting on this gene region both in the Greek Black Pig breed and in many other local pig breeds, that usually have also a quite low average piglet size. However, if we compare the allele frequency at this gene marker with what observed in wild boars (fixed in the wild-type allele; Appendix A), it is evident that this Greek breed has a clear signature of domestication. The same evidence also comes from the allele frequency of the *IGF2* polymorphism, where the mutated allele *A*, associated with the higher growth rate and leaner carcass traits than the alternative allele [30], is the predominant allele in the Greek Black Pig breed. Here, the breed is more similar to a highly selected cosmopolitan breed than other local pig breeds, where allele A is sometime absent or is at very low frequency (Appendix A).

## 5. Conclusions

The investigation of DNA markers in untapped pig breeds can disclose the genetic history of these autochthonous animal genetic resources, usually well adapted to the local production systems. The quite high genetic variability detected in the five genes investigated in the Greek Black Pig breed indicated that admixture with both wild boar populations and other pig breeds occurred quite frequently (probably before the beginning of the conservation program), suggesting that it would be important to establish a well-defined plan able to maintain the genetic integrity of this pig breed. Other studies, including the genotyping of additional candidate gene markers and the analysis of thousands of SNPs covering the whole genome of the animals are needed to obtain a more detailed and improved picture of the two directions of gene flow (from wild boars and from cosmopolitan breeds) that might have shaped the current population genomic structure of the Greek Black Pig breed.

The identification of a novel *MC1R* allele in this breed, not previously reported in any other pig populations, indicates that local animal genetic resources can be the reservoir of interesting genetic variants that can be useful to understand basic biological mechanisms underlying relevant processes and phenotypic traits, with potential applications in other fields, including the use of unique gene markers for local breed product authentication.

## Figures and Tables

**Figure 1 animals-13-01763-f001:**
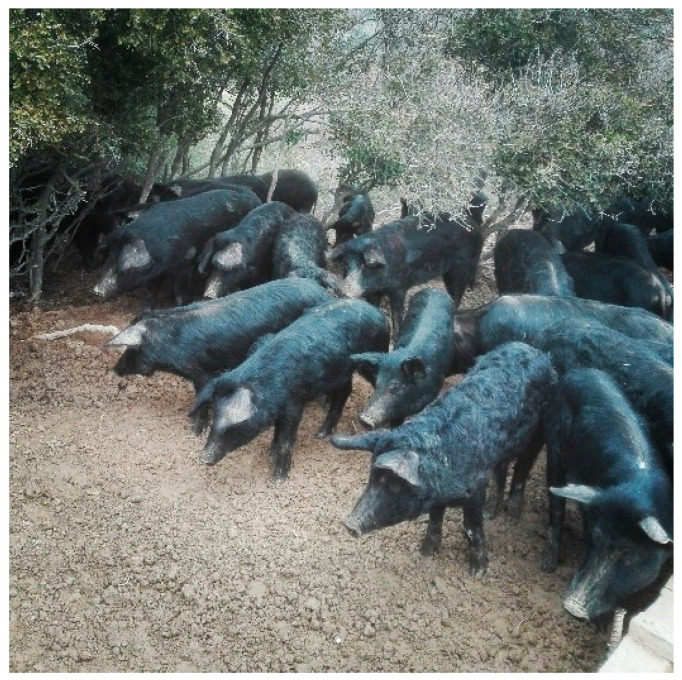
Greek Black Pigs.

**Figure 2 animals-13-01763-f002:**
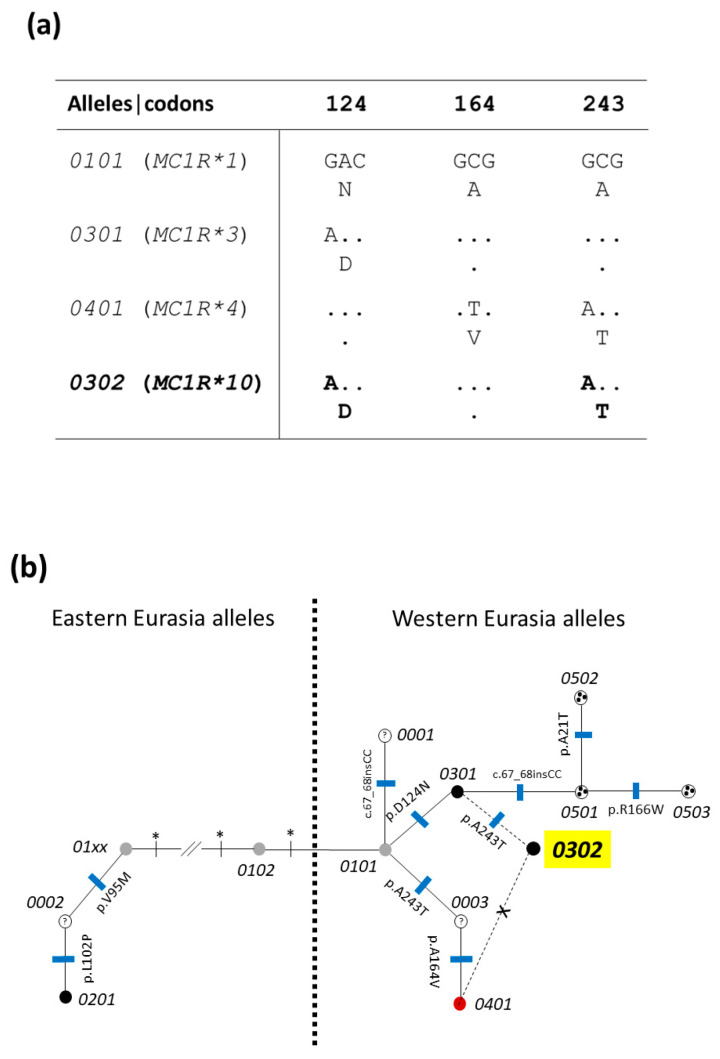
(**a**) Mutations in the *MC1R* gene defining four alleles aligned starting from the sequence of allele *0101* (*MC1R*1*, allele *E^+^*) of European wild boars. The corresponding nucleotides are indicated for allele *0301* (*MC1R*3*, *E^D2^*), allele *0401* (*MC1R*4*, *e*) and the new allele *0302* (*MC1R*10*, *E^D2e^*), outlined in red. The codon numbers are reported at the top of the triplets of the first allele, which encode the corresponding amino acids indicated below the triplets. Dots indicate that the nucleotides or the amino acids are the same as those of the top sequence. (**b**) A simplified median-joining network of porcine *MC1R* alleles modified from previous studies [21,37], including a few Eastern Eurasia (Asian)-derived alleles and all Western Eurasia (European)-derived alleles (indicated as nodes in the tree). Nomenclature of the alleles is based on the four-digit system previously established [21,37], where the first two digits represent coat color and the last two digits distinguish different alleles of the same color type and similar basic sequence. Color of the nodes represent the associated color effect on pig coat of the indicated alleles. Where the association with a coat color is not defined, a white circle with a question mark is reported. Some of these sequences have not been identified yet, but they correspond to intermediate sequences that produced other detected alleles after an additional mutation event occurred. Each branch between nodes represents a single-nucleotide change. Missense mutations are reported. Blue ticks perpendicular to each branch correspond to missense mutations or the two-nucleotide insertion. Asterisks indicate synonymous substitutions that distinguish different *MC1R* sequences. The Eastern Eurasian allele series is not complete: only relevant steps, that lead to allele *0201*, detected in our study, are reported. The new European allele *E^D2e^* or *MC1R*10*, indicated in the figure as *0302*, is outlined in yellow. As the phylogenetic origin of this allele is uncertain, the two possible branches with two potential phylogenetic nodes are dotted. The branch between *0401* and *0302* is derived by a crossing over event (indicated with a cross) and not by a mutation event. Information of the breeds carrying the different alleles are already reported in [9,13,19,20,21,22,37].

**Figure 3 animals-13-01763-f003:**
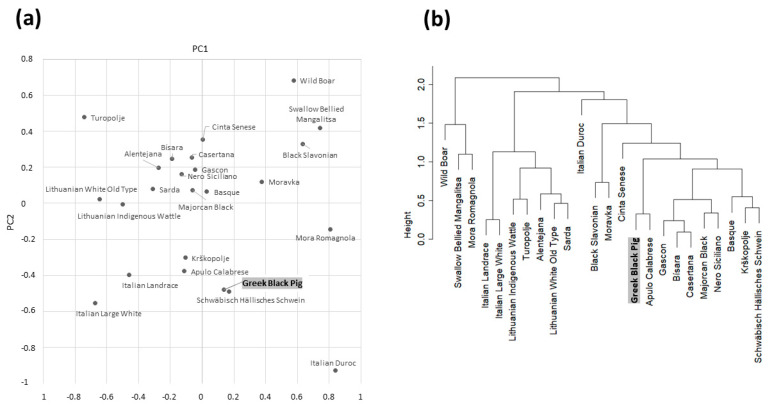
Graphical representations of the genetic distance between the investigated pig breeds and the wild boar population based on *MC1R, KIT, NR6A1*, *VRTN* and *IGF2* allele frequency information. (**a**) Principal Component Analysis (PCA) plot. (**b**) Cluster dendrogram obtained with the Euclidean distance among breeds.

**Table 1 animals-13-01763-t001:** Allele frequencies in the Greek Black Pig breed in the five investigated genes.

Gene	*MC1R*						*KIT * ^1^		*NR6A1 * ^2^		*VRTN * ^2^		*IGF2*	
Allele	*E^+^*	*E^D1^*	*E^D2^*	*E^D2e^*	*E^P^*	*E*	C	T	T	C	Q	wt	G	A
Frequency	0.076	0.009	0.703	0.042	0.017	0.153	0.983	0.017	0.161	0.839	0.449	0.551	0.144	0.856

^1^ Reported allele frequencies are referred to the PCR-RFLP genotyping results of the g.8:43597545C > T SNP. The duplication breakpoint was identified in only one pig out of 59. ^2^ The locus was not in the Hardy–Weinberg equilibrium (*p* < 0.05).

## Data Availability

Gene sequence data have been deposited in the European Nucleotide Archive (ENA) with accession number PRJEB59417.

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
