# Peer review of "Signatures of Admixture and Genetic Uniqueness in the Autochthonous Greek Black Pig Breed Deduced from Gene Polymorphisms Affecting Domestication-Derived Traits"

_animals, 2023, doi:10.3390/ani13111763_

Round 1
Reviewer 1 Report
This work focuses on the genetic variability of rare and small breeds. Increasing the number of works on this topic provides knowledge about the condition of rare breeds and the monitoring of their genetic variability. The presented work analysed samples from the indigenous Greek Black Pig breed. The results were than compared with the results from other studies available.
In the study, the authors selected five genes with a confirmed impact on the swine phenotype (colour, body length, carcass parameters).
The Introduction is written well and comprehensively.
I recommend some modifications.
Materials and Methods
1. The authors aimed to compare the results obtained for the Greek Black Pigs with other Mediterranean and Western Balkan pig breeds. However, in M&M are listed breeds from Germany and Lithuania. These countries are situated in northern Europe. Interestingly there are no samples from the countries located in Asia.
2. Please provide the number of samples in 2.3. subsection. At least the total number of samples from additional 22 pig breeds and one wild boar population should be added. Although it is available in Table S2, it is important enough to give it in the main text.
Results
1. Figure 2 b – which of the breeds listed in 2.3 subsections are from Eastern Eurasia (Asian)?
Discussion
1. Should be added a short note that MC1R belongs to the group of genes affecting skin/coat colour. The discussion about the real coat colour of the breed studied without mentioning that it might result from the effect of other genes (de facto not analysed here) is not comprehensive.
2. The Discussion does not mention Other genes (markers) studied.
Author Response
Reviewer 1
This work focuses on the genetic variability of rare and small breeds. Increasing the number of works on this topic provides knowledge about the condition of rare breeds and the monitoring of their genetic variability. The presented work analysed samples from the indigenous Greek Black Pig breed. The results were than compared with the results from other studies available.
In the study, the authors selected five genes with a confirmed impact on the swine phenotype (colour, body length, carcass parameters).
The Introduction is written well and comprehensively.
AUTHORS
Thank you for the comments.
-----------------------------------------
Reviewer 1
I recommend some modifications.
Materials and Methods
- The authors aimed to compare the results obtained for the Greek Black Pigs with other Mediterranean and Western Balkan pig breeds. However, in M&M are listed breeds from Germany and Lithuania. These countries are situated in northern Europe. Interestingly there are no samples from the countries located in Asia.
AUTHORS
Thank you for the comments.
We compared Greek Black Pig results with the autochthonous European pig breeds in almost all areas of Europe: some Mediterranean breeds (Italian, French, Spanish and Portuguese ones), some Northern European breeds (including one from Germany and one from Lithuania) and other Balkan breeds, near to Greece. We decided to compare only among Europe because the aim of this study was to characterize Greek Black Pig and their interaction with other European pig population, including wild boars. Moreover, we wanted to use only data that were derived from our studies and not from the studies published by other that however would have been difficult to be compared, even for just few markers.
--------------------------------------------
Reviewer 1
- Please provide the number of samples in 2.3. subsection. At least the total number of samples from additional 22 pig breeds and one wild boar population should be added. Although it is available in Table S2, it is important enough to give it in the main text.
AUTHORS
Thank you for the comment. We added this information.
--------------------------------------------
Reviewer 1
Results
- Figure 2 b – which of the breeds listed in 2.3 subsections are from Eastern Eurasia (Asian)?
AUTHORS
Figure 2b is a graphical representation of a median-joining network of all MC1R alleles already described in the literature in order to show the possible relationship of the new allele that we detected in MC1R in Greek Black pigs with other MC1R alleles detected in many other pig populations. We built this network modifying it from previous studies (Fang et al., 2009, Linderholm et al., 2016) and adapted it according to our hypotheses of origin of this new variant.
For this reason, there is no direct correlations between the breeds that we analyzed to compare the Greek Black Pig genotyping results to the different MC1R alleles reported in the median-joining network. Information on the breeds that carries the different alleles has been already reported in the cited references and summarized in the review of Fontanesi (2022). To better clarify this issue, we included a sentence in the legend of figure 2 to refer to these works for more details.
--------------------------------------------
Reviewer 1
Discussion
- Should be added a short note that MC1Rbelongs to the group of genes affecting skin/coat colour. The discussion about the real coat colour of the breed studied without mentioning that it might result from the effect of other genes (de factonot analysed here) is not comprehensive.
AUTHORS
Thank you for the suggestion. Since we dedicated a long paragraph in the Introduction (lines 88-101) on MC1R gene and its role on the coat colour phenotypes and its breed associated alleles, in the Discussion section we focused mainly on the derived explanations of our results related to the new variant detected and the probable introgression of Greek Black Pig breed with wild boars or other pig populations.
--------------------------------------------
Reviewer 1
- The Discussion does not mention Other genes (markers) studied.
AUTHORS
Thank you for the suggestion. We have integrated the discussion with a part focused on VRTN and IGF2 as the first part of the discussion already discussed the results derived from NR6A1 and KIT gene markers.
Reviewer 2 Report
The authors reported a novel MC1R allele in a local pig breed, however, it is difficult to evaluate whether finding MC1R allele, relating coat color, is important in this breed. Authors had better to show why it is worthy to publish this results.
Author Response
Reviewer 2
The authors reported a novel MC1R allele in a local pig breed, however, it is difficult to evaluate whether finding MC1R allele, relating coat color, is important in this breed. Authors had better to show why it is worthy to publish this results.
AUTHORS
Thank you for the comments.
The main aim of this study was to characterize for the first time in the local Greek Black Pig breed some important polymorphisms in five genes whose variability affects coat colour, teat number and meat production traits. Those genetic markers are important, also, because usually their allele serie include breed-specific alleles, in particular at the MC1R gene, associated with typical coat colour of different pig populations. The main results we obtained are that i) in all gene investigated Greek Black Pig breed shows no allele fixation, meaning that introgression events occurred between this breed and other pig breeds, including wild boars (wild type alleles have been identified in MC1R and NR6A1 gene, indicating active gene flow with wild boars) and ii) the novel MC1R allele carried by 3 animals never detected before, indicates a genetic uniqueness of Greek Black Pig breed. Moreover, this demonstrated that autochthonous local breeds can be interesting reservoir of unexpected genetic variants and thus it is important to develop breeding programs to preserve these genetic resources.
We extensively and clearly reported the value of the results obtained in the MC1R gene starting from a detailed description of the state of the art in the Introduction, a detailed description of the results and an extended discussion. The value of the information reported in the MC1R gene is also very well established in the cited literature.
The identification of a new allele at this gene opened interesting perspectives for additional studies reported in the conclusions.
Reviewer 3 Report
In this article titled “Signatures of De-Domestication, Admixture and Genetic Uniqueness in the Autochthonous Greek Black Pig Breed Deduced from Gene Polymorphisms Affecting Domestication-Derived Traits”, Ribani and co-workers performed a genetic characterization of a Greek local pig breed, Greek Black Pig, by analysing some gene markers associated with morphological and production traits, identifying also a novel allele at MC1R gene.
The research takes place in an interesting context and could be useful for maintaining a local breed, protecting it from genetic pollution by wild boar or other domestic forms of pigs.
In general the article is written in quite clear and fluent language. However, throughout the manuscript I found too general sentences, often without any reference, which in my opinion should be better explained.
Furthermore, it’s necessary clarify better what the author mean with “de-domestication”.
Below are some of my comments:
Line 28: it would be better write also the scientific name of the species, at least the first time that the authors cite it.
Line 49-52: The authors could report a reference for this.
Line 61 and line 63: “…is an autochthonous breed raised in Greece”…this sentence is repeated in the two lines. Rephrase.
Line 65: What the authors would mean with “traditional outdoor livestock production”. This could be clarify. There are various way to heard pigs outdoor and this could affect your definition of de-domestication (see also after).
Line 66: What the authors would mean with “harsh environment”, could they give a better description of this?
Line 69-70: The authors could report reference for this statement.
Line 77: “.at birth […, remove dot before 80%.
line 108-110: I understand that the authors focus specifically on their species but they should provide more insight into what they report. Some sentences seem too generic and without references. Here, for example, the authors could report some other characters involved in the domestication process such as floppy ears (https://doi.org/10.1371/journal.pgen.100345), reduction of the sense of smell (https://doi.org/10.1007/s11692-013-9262-3), decrease in the ratio of brain weight to body weight (https://doi.org/10.1093/gbe/evx186), curled tail.
line 114-115: Also for this statement the authors should report a reference?
Line 117: What do the authors mean by “carcass parameters”? Please specify.
142: I would suggest calling this paragraph something like "sampling of Greek Black Pigs". Otherwise, leave this title and move all the description referring to the species here
line 144: The authors should insert a figure of the study area (with clear geographical references) or at least report the geographical coordinates of the investigated areas in order to allow even readers unfamiliar with these areas to understand the context in which the research is conducted.
line 146: The authors here should be more specific. What is meant by adults? Could they refer to age, weight, size? on the basis of what characteristics they determined that pigs were adults?
lines 147-148: “…with the typical morphological traits of the breed”.
Do the authors refer to the medium length ears mentioned above or are there also other features? If yes, as it appears from what is written here, what are these other traits?
lines 150-151: Did the authors use controls during the extraction and evaluate the quality of the extracted DNA? All DNAs were successfully extracted? These details should be reported.
lines 320-322: Maybe this should be moved up for discussion.
line 332: several genetic events…for example?
lines 340-342: The return of wild type traits (therefore typical of the wild form) in a domesticated form that had lost those traits due to domestication, could also be connected to the outdoor lifestyle in which these animals are breeding.
There is a large literature on this subject and the process is called feralization (i.e.https:// doi.org/10.1038/s41467-020-14515-6). This process has been studied in various geographical areas (Sardinia, France, southern Italy, etc.) and affects not only pigs (https://doi.org/10.1111/mec.16238) but also others such as horses (https://link.springer.com/chapter/10.1007/978-0-306-48215-1_1), sheep (https://doi.org/10.1111/j.1469-7998.2008.00531.x), chickens (https:// doi.org/10.1071/ZO9910439).
Feralisation in pigs takes place via free grazing practices or farming practices such as pig transhumance (https://core.ac.uk/download/pdf/83940349.pdf). Also for this is important to clarify in this manuscript the way of breeding of Greek pig.
Can Geek pig be considered like-feral?
I don't know if the two processes (feralization and de-domestication) are the same thing for you, but if not, however, it would make sense to mention feralization as a widely described phenomenon and highlight the elements that differentiate feralization and de-domestication.
So, I suggest to change the title by removing reference to “de-domestication”.
358-360: I understand that the authors are experts in this field however they could also expand their references by citing other sources that corroborate their claims and not just auto-cite.
Line 361-363: Could the authors here provide a quote to say that the Greek Black Pig is more adapted to harsh environments than other pig breeds? So, the Greek pig live in wild?
lines 438-439: This sentence is too general. Authors should better clarify what they mean by “…relevant processes and phenotypic traits..” and also “…with potential applications in other fields”.
Author Response
Reviewer 3
In this article titled “Signatures of De-Domestication, Admixture and Genetic Uniqueness in the Autochthonous Greek Black Pig Breed Deduced from Gene Polymorphisms Affecting Domestication-Derived Traits”, Ribani and co-workers performed a genetic characterization of a Greek local pig breed, Greek Black Pig, by analysing some gene markers associated with morphological and production traits, identifying also a novel allele at MC1R gene.
The research takes place in an interesting context and could be useful for maintaining a local breed, protecting it from genetic pollution by wild boar or other domestic forms of pigs.
In general the article is written in quite clear and fluent language. However, throughout the manuscript I found too general sentences, often without any reference, which in my opinion should be better explained.
AUTHORS
Thank you for the positive comments. We changed the text and rephrased sentences in some points, as you suggested.
--------------------------------------------
Reviewer 3
Furthermore, it’s necessary clarify better what the author mean with “de-domestication”.
AUTHORS
With “De-domestication” we mean the reverse process of domestication, in which the human artificial directional selection is low and wild type alleles are free to re-enter into the population. We found out this process in the Greek Black Pig breed in some of the investigated genes, in which some alleles typical of wild boars (that are almost fixed in wild boars but usually absent in domestic populations) have been detected with a quite high frequency. This could have negative impacts on productive performances and of course on the genetic integrity of the local breed. The concept is different from feralization where animals are not managed at all and free to mate without any restriction with the wild counterpart, when it is present in the same region/land.
--------------------------------------------
Reviewer 3
Below are some of my comments:
Line 28: it would be better write also the scientific name of the species, at least the first time that the authors cite it.
AUTHORS
Thank you for the suggestion. In the simple summary, for the definition, we wanted to keep simple the text. The information related to the species is already included in the keywords. Sus scrofa is also included in the first sentence of the Introduction.
--------------------------------------------
Reviewer 3
Line 49-52: The authors could report a reference for this.
AUTHORS
References for this little generic section are included among references from #1 to #4, especially for Larson and colleagues, 2005 (Science). We have put the Larson at al. reference at the end of the sentence which ends in line 52.
--------------------------------------------
Reviewer 3
Line 61 and line 63: “…is an autochthonous breed raised in Greece”…this sentence is repeated in the two lines. Rephrase.
AUTHORS
Thank you for the suggestion, we rephrased the sentences.
--------------------------------------------
Reviewer 3
Line 65: What the authors would mean with “traditional outdoor livestock production”. This could be clarify. There are various way to heard pigs outdoor and this could affect your definition of de-domestication (see also after).
AUTHORS
We mean extensive and semi-extensive breeding, we added it in the text.
--------------------------------------------
Reviewer 3
Line 66: What the authors would mean with “harsh environment”, could they give a better description of this?
AUTHORS
With “harsh” we mean the typical Mediterranean dry, rough and hot environmental conditions of Greece, mainly in the summer. There are no forests, but mainly a semi-desert countryside in Greece, and these are no favourable conditions for livestock if they are not well adapted to environment.
--------------------------------------------
Reviewer 3
Line 69-70: The authors could report reference for this statement.
AUTHORS
Thank you for the suggestion, we added a reference at the end of the sentence
--------------------------------------------
Reviewer 3
Line 77: “.at birth […, remove dot before 80%.
AUTHORS
Thank you for the suggestion, we removed the dot.
--------------------------------------------
Reviewer 3
line 108-110: I understand that the authors focus specifically on their species but they should provide more insight into what they report. Some sentences seem too generic and without references. Here, for example, the authors could report some other characters involved in the domestication process such as floppy ears (https://doi.org/10.1371/journal.pgen.100345), reduction of the sense of smell (https://doi.org/10.1007/s11692-013-9262-3), decrease in the ratio of brain weight to body weight (https://doi.org/10.1093/gbe/evx186), curled tail.
AUTHORS
The study was mainly focused on the Greek Black Pig breed.
We decided to focus on the five genes investigated in this work because they are related to specific phenotypes relevant not only for the for the domestication process (which is not the main aim of this study) but also for the genetic characterization of this breed. Indeed, the genetic diversity of Greek Black Pig breed has never investigated before as well as its genetic integrity; as far as we know, several introgression events might have occurred during the past years, but they have been unexplored. These genetic markers are also useful to monitor these different events because wild boars specific alleles in these genes are detectable in pig breeds and could indicate a genetic flow between domestic and wild pigs. In order to preserve a local breed, the first step is to characterize the allelic frequencies of some candidate genes associated to relevant phenotypes, trying also to detect some breed specific allele.
In fact, coat colour phenotypes and their related genes are one of the most important traits to characterize and investigate for developing an appropriate conservation program of a local pig breed, together with other important productive traits such as body length (associated also to reproductive performances) or meat quality.
On the other hand, we have already explored some traits related to pig domestication in several populations, for instance the reduction of bitter taste perception (https://onlinelibrary.wiley.com/doi/epdf/10.1111/age.12472), but again it was not the aim of our study.
--------------------------------------------
Reviewer 3
line 114-115: Also for this statement the authors should report a reference?
AUTHORS
Thank you for the suggestion, we added a reference.
--------------------------------------------
Reviewer 3
Line 117: What do the authors mean by “carcass parameters”? Please specify.
AUTHORS
With “carcass parameters” we mean phenotypes related to meat quality and carcass quality for meat production.
--------------------------------------------
Reviewer 3
142: I would suggest calling this paragraph something like "sampling of Greek Black Pigs". Otherwise, leave this title and move all the description referring to the species here
AUTHORS
Thank you for the suggestion, we change the title of the paragraph with “Greek Black Pig Samples”.
--------------------------------------------
Reviewer 3
line 144: The authors should insert a figure of the study area (with clear geographical references) or at least report the geographical coordinates of the investigated areas in order to allow even readers unfamiliar with these areas to understand the context in which the research is conducted.
AUTHORS
Thank you for the comment, we added the GPS coordinates of the sampling area in the text
--------------------------------------------
Reviewer 3
line 146: The authors here should be more specific. What is meant by adults? Could they refer to age, weight, size? on the basis of what characteristics they determined that pigs were adults?
AUTHORS
With “adults” we refer to pigs, sows and boars, reproductively active. They were not piglets or young pigs or gilts.
--------------------------------------------
Reviewer 3
lines 147-148: “…with the typical morphological traits of the breed”.
Do the authors refer to the medium length ears mentioned above or are there also other features? If yes, as it appears from what is written here, what are these other traits?
AUTHORS
Exactly, we mean the black coat colour, the body length and morphology. The description of the breed is reported in the DAD-IS database of the FAO that is cited in the text.
--------------------------------------------
Reviewer 3
lines 150-151: Did the authors use controls during the extraction and evaluate the quality of the extracted DNA? All DNAs were successfully extracted? These details should be reported.
AUTHORS
Yes, we always use controls for DNA extraction efficiency and we extracted DNA in double aliquots, in order to confirm results at each steps. After DNA extraction we performed quality check using a nano spectrophotometer instrument and running DNA samples in a 1% agarose electrophoresis gel. All samples were successfully extracted and then amplified. Also for PCR amplifications we used negative and positive controls as well as for genotyping. In addition, we performed Sanger sequencing also to confirm genotyping data. We did not add these steps in the text because these are routinary steps for us to confirm results. These details are not reported any more in scientific papers that describe similar analyses as we did as they are part of the laboratory routines.
--------------------------------------------
Reviewer 3
lines 320-322: Maybe this should be moved up for discussion.
AUTHORS
Thank you for the comment. In lines 320-322 we described the graphical representations of genetic distance between Greek Black Pig breed and the other pig populations (Fig. 3a and 3b). We reported the concept in lines 334-343, in which even if we found some wild type alleles in this local breed it does not mean that Greek Pigs are genetically closed to wild boars, but rather that several introgression events with wild boars may occurred in this breed. <these sentences are descriptive parts of the results, in comparisons with the information from other breeds.
--------------------------------------------
Reviewer 3
line 332: several genetic events…for example?
AUTHORS
We expanded this concept in the next lines: with “several genetic events” we mean basically i) the genetic flow with wild boars due to the extensive and semi-extensive breeding system of Greek Black Pigs and ii) the genetic admixture with other pig breeds that occurred during the reconstruction of this local breed after a severe reduction of its effective population in the 1960s.
--------------------------------------------
Reviewer 3
lines 340-342: The return of wild type traits (therefore typical of the wild form) in a domesticated form that had lost those traits due to domestication, could also be connected to the outdoor lifestyle in which these animals are breeding.
There is a large literature on this subject and the process is called feralization (i.e. https:// doi.org/10.1038/s41467-020-14515-6). This process has been studied in various geographical areas (Sardinia, France, southern Italy, etc.) and affects not only pigs ( https://doi.org/10.1111/mec.16238 ) but also others such as horses (https://link.springer.com/chapter/10.1007/978-0-306-48215-1_1), sheep ( https://doi.org/10.1111/j.1469-7998.2008.00531.x ), chickens (https:// doi.org/10.1071/ZO9910439).
Feralisation in pigs takes place via free grazing practices or farming practices such as pig transhumance (https://core.ac.uk/download/pdf/83940349.pdf). Also for this is important to clarify in this manuscript the way of breeding of Greek pig.
Can Greek pig be considered like-feral?
I don't know if the two processes (feralization and de-domestication) are the same thing for you, but if not, however, it would make sense to mention feralization as a widely described phenomenon and highlight the elements that differentiate feralization and de-domestication.
So, I suggest to change the title by removing reference to “de-domestication”.
AUTHORS
Thank you for the comment. We are aware about concepts of feralization and de-domestication. The two terms are difficult to separate completely, and it would be also very difficult to demonstrate that, without any doubts, there are some signatures of de-domestication or feralization in the Greek Black Pig breed population based on the simple study we have carried out. Therefore, as suggested, we have removed the word de-domestication from the title and added a few elements as disclaimer in the Discussion where the de-domestication concept was originally introduced and that now we also mention feralization. A few other references were cited here.
--------------------------------------------
Reviewer 3
358-360: I understand that the authors are experts in this field however they could also expand their references by citing other sources that corroborate their claims and not just auto-cite.
AUTHORS
Thank you for the comment. We refer also to literature therein our previous studies.
--------------------------------------------
Reviewer 3
Line 361-363: Could the authors here provide a quote to say that the Greek Black Pig is more adapted to harsh environments than other pig breeds? So, the Greek pig live in wild?
AUTHORS
Actually, they live free-range in the Greek Mediterranean environment, in which there are also semi-desert conditions.
--------------------------------------------
Reviewer 3
lines 438-439: This sentence is too general. Authors should better clarify what they mean by “…relevant processes and phenotypic traits..” and also “…with potential applications in other fields”.
AUTHORS
Thank you for the suggestions, we added some example in the text.
Round 2
Reviewer 1 Report
The revision is fairly well-made and the comments were addressed adequately. I have no more comments.
Reviewer 2 Report
Accepted.
Reviewer 3 Report
The authors have resolved almost all of my requests and I consider the matter resolved, although I do not yet have an unambiguous description of how these animals are bred and the extent of contact with the wild, and therefore the difference between de-domestication and feralization is not clear to me (a very thin border).